# Alignment-guided Temporal Attention for Video Action Recognition

**Yizhou Zhao**[1*†]     **Zhenyang Li**[2*†]     **Xun Guo**[3]     **Yan Lu**[3]
[1]Carnegie Mellon University    [2]Tsinghua University    [3]Microsoft Research Asia
yizhouz@andrew.cmu.edu, lizy20@mails.tsinghua.edu.cn
{xunguo, yanlu}@microsoft.com

## Abstract

Temporal modeling is crucial for various video learning tasks. Most recent approaches employ either factorized (2D+1D) or joint (3D) spatial-temporal operations to extract temporal contexts from the input frames. While the former is more efficient in computation, the latter often obtains better performance. In this paper, we attribute this to a dilemma between the sufficiency and the efficiency of interactions among various positions in different frames. These interactions affect the extraction of task-relevant information shared among frames. To resolve this issue, we prove that frame-by-frame alignments have the potential to increase the mutual information between frame representations, thereby including more task-relevant information to boost effectiveness. Then we propose Alignment-guided Temporal Attention (ATA) to extend 1-dimensional temporal attention with parameter-free patch-level alignments between neighboring frames. It can act as a general plug-in for image backbones to conduct the action recognition task without any model-specific design. Extensive experiments on multiple benchmarks demonstrate the superiority and generality of our module.

## 1 Introduction

Unlike images, videos include a wealth of temporal information due to common and distinct patterns in nearby frames. In images, the whole scene is static. In videos, however, the background, the foreground, and each part of them can have different degrees of movement. These differences necessitate temporal modeling in video learning tasks.

With respect to this, recent literature mainly focuses on two branches of modelings, namely, factorized (2D+1D) spatial-temporal operations [49, 15, 3, 34] and joint (3D) spatial-temporal operations [42, 11, 37, 32]. The former considers staggering or dividing spatial operations with temporal ones, such as separable 3D convolutions and factorized spatial-temporal attention. Since the two groups of operations are relatively independent, this paradigm leverages indirect interactions for features at different space-time locations. Enjoying the efficiency from fewer participants for each operation, it often suffers from less sufficiency of cross-frame cross-location interactions. In contrast, the latter extends 2-dimensional operations to their 3-dimensional varieties, thereby jointly interacting features from neighboring areas as well as frames, e.g., 2D convolutions vs. 3D convolutions and spatial attention vs. joint spatial-temporal attention. This line of work explores a more global view compared with the former one, bringing more direct and adequate interactions at the cost of dramatically increased complexity. As a result, a conflict seemingly exists between the sufficiency and the efficiency of interactions among various positions in different frames.

---

[*]Work done during internship at Microsoft Research Asia.
[†]Equal contribution.

36th Conference on Neural Information Processing Systems (NeurIPS 2022).

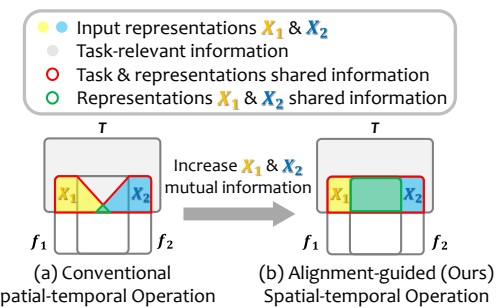

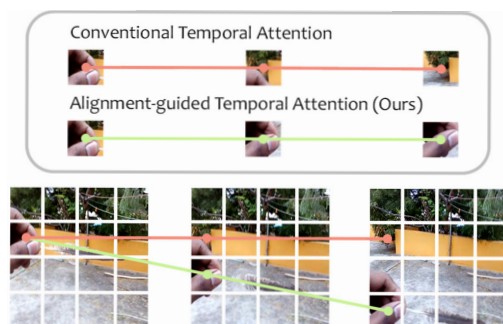

Figure 1: **Information diagram illustrating our motivation.** $T$ stands for task-relevant information, and $X_1$, $X_2$ denote representations of adjacent input frames $f_1$, $f_2$.

Figure 2: **Comparison between conventional Temporal Attention and Alignment-guided Temporal Attention (ATA).** Participants of each operation are connected with lines.

To address this conflict, we revisit both schemes from the perspective of information theory. With the help of more direct interactions, joint spatial-temporal operations share more mutual information between neighboring frame representations than their factorized counterparts. This potentially increases task-relevant information extracted by the model, benefiting its performance. Nevertheless, for joint spatial-temporal modeling, the ability to exploit mutual information from consecutive frames depends on the kernel size of convolution-based operations or the window size of attention-based ones. To reach the theoretical upper bound of mutual information shared, completely global interactions are needed for this paradigm, which is not realistic. In general cases, these conventional schemes collect inadequate task-relevant information as illustrated in Fig. 1(a).

Alternatively, we propose to directly interact between the most similar parts of adjacent frames, instead of among regions at nearby spatial locations. In other words, we guide the spatial-temporal operation with alignments as shown in Fig. 2. We theoretically prove that this guidance increases mutual information of neighboring frame representations as in Fig. 1(b) and practically implement this concept as our *Alignment-guided Temporal Attention* (ATA). Upon traditional Temporal Attention, it has no extra parameters and low additional computation cost. For alignment, we match patches of adjacent frames with the Kuhn-Munkres Algorithm (KMA) [4] based on cosine similarity. After temporal attention through the aligned route, we de-align patches to keep their original spatial order.

Our contributions can be summed up as follows.

1. We discuss the problems of factorized (2D+1D) and joint (3D) spatial-temporal operations and propose Alignment-guided Temporal Attention (ATA) for temporal modeling. Taking advantages of both, ATA achieves strong performance while maintaining low computational overheads.

2. We theoretically prove that the frame-by-frame alignment increases mutual information between neighboring frame representations, thereby potentially including more task-relevant information.

3. Through qualitative and quantitative experiments, we demonstrate that our ATA is superior to conventional temporal modeling methods by providing more effective cross-frame cross-location interactions. Implementations on video frameworks and image backbones (TimeSformer [3], CycleMLP [6], and ConvNeXt [31]) show its potential as a general plug-and-play module to bridge from image recognition to video action recognition.

## 2 Related Work

**Visual learning architectures.** Due to the high generalization of ViT [10], numerous Transformer-based image backbones recently extend their work range to video. TimeSformer [3] presents four kinds of factorized space-time implements utilizing the standard ViT structure. VTN [33] uses a temporal attention model to aggregate information from all video clips. MViT [11] creates a multi-scale architecture through integrating pooling and downscale methods, which have both been used in image backbones. ViViT [2] creates three variants that focus on factorized space-time attention. It leverages tubelet embeddings to extend frame patch representations into spatial-temporal volume representations.

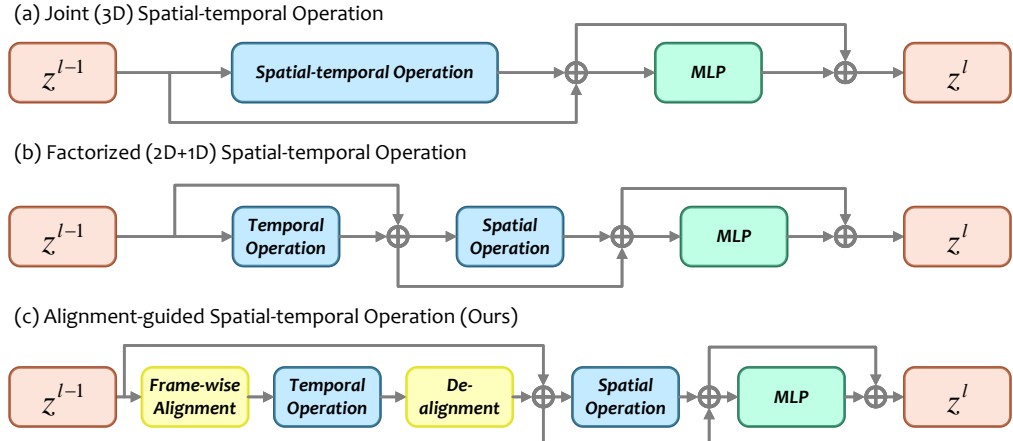

Figure 3: **Comparison among spatial-temporal operations in terms of model structures.**

**Video Action recognition.** Early action recognition research [26, 43, 35, 47, 13] focuses on hand-crafted features. With the advancement of deep neural networks, schemes adopting learnable video representations gradually surpass traditional methods based on DT [44] and iDT [43]. To be specific, AlexNet [23] generates numerous important video architectures, which include 2D/3D CNN [21, 1, 28] in two-stream models and its variants with LSTM/GRU/RNNs [9, 38, 51]. Recently, Transformers-based ViT [10] spawns a series of video learning architectures on the basis of the attention mechanism, such as VTN [33], TimeSformer [3], ViViT [2], and Video Swin [32].

**Temporal correspondences.** In comparison to image recognition, motion representation is the most difficult aspect of video action recognition [25, 20, 36, 18, 34, 48]. Recent alignment methods have primarily focused on supervised temporal correspondences. For representing temporal relationships, optical flow [16, 17, 27, 39–41] is proposed to establish the displacements in pixels between successive frames. Patch-level interconnections can reduce the memory cost of optical flow. For instance, [29] presents a typical descriptor model called SIFT flow and the SIFT features are being used for matching patches. Local features such as HOG [24, 7] and MBH [8] can be used to describe statistical features as well. The correlation of patches or other inter-frame components, in particular, has positive effects on models learning visual representations, according to [41].

## 3 Method

For video learning tasks, we consider their inputs a combination of spatial and temporal information. The former is introduced by sequential static frames, while the latter originates from their potential connections. These two parts are usually tackled jointly in convolution-based methods [5] as well as in attention-based approaches [30], with their per-block model structures roughly demonstrated in Fig. 3(a), i.e., joint (3D) spatial-temporal operations. We also notice attempts [49, 2, 3] separating spatial operations from temporal ones as in Fig. 3(b), known as factorized (2D+1D) spatial-temporal operations. For either scheme, the ability to extract mutual information from consecutive frames is restricted by the kernel/window size of its temporal operation. To mitigate this, we propose to conduct frame-wise alignment before each temporal operation and de-alignment after that as in Fig. 3(c).

### 3.1 Problem definition

To begin with, we define alignment generally. Given two neighboring frames $X^{t-1}, X^t \in \mathbb{R}^{HW \times C}$ with their patch embeddings $\left\{x_i^{t-1}\right\}_{i=1}^{HW}, \left\{x_j^t\right\}_{j=1}^{HW}$, the alignment function of $X^{t-1}$ and $X^t$ is

$$\alpha\left(X^{t-1}, X^t\right) = \arg\max_{\rho}\left\{\chi\left[X^{t-1}, \rho(X^t)\right]\right\} \tag{1}$$

where $\chi$ is a criterion function and $\rho : \mathbb{R}^{HW \times C} \to \mathbb{R}^{HW \times C}$ is a projection. The de-alignment function $\alpha^{-1}$ is defined as the reverse process of alignment

$$\alpha^{-1}\left(\alpha\left(X^{t-1}, X^t\right)\right) = X^t \tag{2}$$

Specifically, we can have spatial rearrangement as a series of projection $\rho$

$$\rho^{\text{Rearrange}}(X^t) = R \times X^t = \hat{X}^t = \{\hat{x}_j^t\}_{j=1}^{HW} \tag{3}$$

where the rearrangement matrix

$$R = \{r_{i,j}^{t-1,t}\} \in \{0,1\}^{HW \times HW}$$
$$\text{s.t.} \sum_{i=1}^{HW} r_{i,j}^{t-1,t} = 1, \sum_{j=1}^{HW} r_{i,j}^{t-1,t} = 1 \tag{4}$$

and cosine similarity as the criterion $\chi$

$$\chi^{\text{Cosine}}\left(X^{t-1}, \hat{X}^t\right) = \sum_{i=j} \frac{x_i^{t-1} \cdot \hat{x}_j^t}{\|x_i^{t-1}\| \|\hat{x}_j^t\|} \tag{5}$$

In this case, the rearrangement can be regarded as a patch-wise match between $X^{t-1}$ and $X^t$, and the alignment should be a perfect matching scheme. As a result, the alignment matrix is one of the rearrangement matrices satisfying

$$A = \arg\max_R \left[\chi^{\text{Cosine}}\left(X^{t-1}, R \times X^t\right)\right] \tag{6}$$

and the de-alignment matrix is simply its transpose

$$D = A^{\text{T}} \tag{7}$$

## 3.2 Proof and analysis

Mutual information of two random variables measures the information shared between them. It represents the degree of reduction in uncertainty of one variable given the other. Traditional 1D temporal operations interact features of consecutive frames at a certain spatial position. Since these features might diverge greatly, their irrelevances can lead to uncertainty, decreasing mutual information collected by the model. We will prove that the mutual information of adjacent frames increases after alignment, which also indicates that the similarity of adjacent frames rises.

We define the representations of two adjacent frames as random variables $X^{t-1}$ and $X^t$. Within the same video clip, they obey the same distribution. Thus their mutual information can be calculated as

$$\text{MI}(X^{t-1}, X^t) = \text{H}(X^t) - \text{H}(X^t \mid X^{t-1}) \tag{8}$$

Eq. (8) expresses the mutual information between the feature $X^{t-1}$ of the $(t-1)^{\text{th}}$ frame and the feature $X^t$ of the $t^{\text{th}}$ frame. The definitions of information entropy $\text{H}(X^t)$ ($t \in \{1, \ldots, T\}$) and conditional entropy $\text{H}(X^t|X^{t-1})$ are as follows.

$$\text{H}(X^t) = -\sum_{j=1}^{HW} p\left(x_j^t\right) \log p\left(x_j^t\right) \tag{9}$$

$$\text{H}(X^t \mid X^{t-1}) = \sum_{x_i^{t-1}} p(x_i^{t-1}) \text{H}(X^t \mid X^{t-1} = x_i^{t-1})$$
$$= -\sum_{x_i^{t-1}} p(x_i^{t-1}) \sum_{x_j^t} p(x_j^t \mid x_i^{t-1}) \log(p(x_j^t \mid x_i^{t-1})) \tag{10}$$

where $x_i^{t-1} \in F^{t-1}$ and $x_j^t \in F^t$. $F^{t-1}$ and $F^t$ denote the feature sets of all patches in the $(t-1)^{\text{th}}$ frame and the $t^{\text{th}}$ frame, respectively. $p(x_i^{t-1})$ is the probability of occurrence of random variable $X^{t-1}$. $p(x_j^t \mid x_i^{t-1})$ is the conditional probability of occurrence of random variable $X^t$ under the condition determined by random variable $X^{t-1}$. $p(x_j^t, x_i^{t-1})$ is the joint probability that random variables $X^{t-1}$ and $X^t$ satisfy certain conditions at the same time.

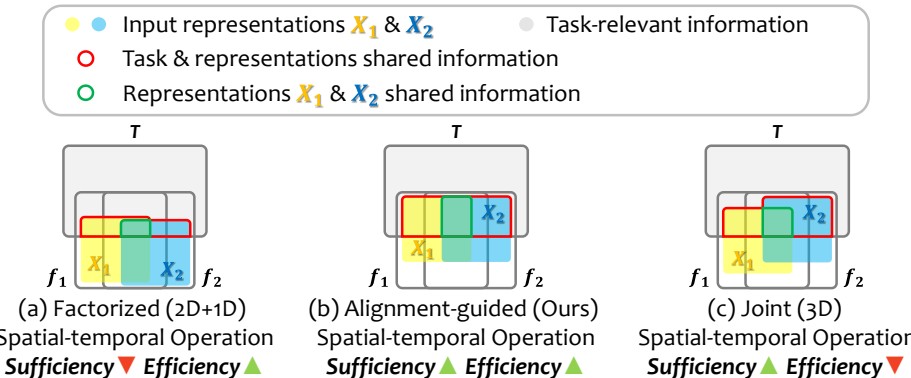

Figure 4: **Comparison of spatial-temporal operations in terms of information diagrams.** $T$ represents task-relevant information, and $X_1$, $X_2$ are representations of adjacent input frames $f_1$, $f_2$.

Given patch embeddings of a certain frame, the probability of a certain patch embedding equaling a certain vector is determined, i.e.,

$$p\left(x_j^t\right) = p\left(\hat{x}_j^t\right) = \frac{1}{HW} \tag{11}$$

where $x_j^t$ indicates the feature of the $j^{th}$ patch in the $t^{th}$ frame before the alignment, and $\hat{x}_j^t$ means the feature of the $j^{th}$ patch in the $t^{th}$ frame after the alignment.

Hence Eq. (10) can be simplified to

$$\mathrm{H}(X^t \mid X^{t-1}) = -\frac{1}{HW} \sum_{x_i^{t-1}, x_j^t} p(x_j^t \mid x_i^{t-1}) \log(p(x_j^t \mid x_i^{t-1})) \tag{12}$$

The rearrangement-based alignment does not change the feature contents of the patch, thus the information entropy does not change after alignment. As a result, the mutual information of the features of two adjacent frames after the alignment is

$$\mathrm{MI}(X^{t-1}, \hat{X}^t) = \mathrm{H}(\hat{X}^t) - \mathrm{H}(\hat{X}^t \mid X^{t-1}) = \mathrm{H}(X^t) - \mathrm{H}(\hat{X}^t \mid X^{t-1}) \tag{13}$$

We notice that the difference between Eq. (8) and Eq. (13) lies in the different conditional entropy $\mathrm{H}(X^t \mid X^{t-1})$ and $\mathrm{H}(\hat{X}^t \mid X^{t-1})$. With $\mathrm{H}(X^t \mid X^{t-1})$ expressed in Eq. (12), $\mathrm{H}(\hat{X}^t \mid X^{t-1})$ can be similarly written as

$$\mathrm{H}(\hat{X}^t \mid X^{t-1}) = -\frac{1}{HW} \sum_{x_i^{t-1}, \hat{x}_j^t} p(\hat{x}_j^t \mid x_i^{t-1}) \log(p(\hat{x}_j^t \mid x_i^{t-1})) \tag{14}$$

We assume that two patches that are close to each other have a greater probability to be similar than the probability to be dissimilar. If we align the patches according to the Eq. (6), more patches with high similarity have the chance to be put in close positions. In this circumstance, patches with high similarity will have a greater probability of appearing at temporally close positions in adjacent frames. On the one hand, when the positions of $i$ and $j$ are close, the probability of similar features appearing in such close locations becomes higher. On the other hand, when the positions of $i$ and $j$ are far away, the dissimilar features are more likely to occur at such a distance too. In general, the conditional probability $p(\hat{x}_j^t \mid x_i^{t-1})$ will become greater than $p(x_j^t \mid x_i^{t-1})$. Consequently, Eq. (10) is greater than Eq. (14)

$$\mathrm{H}(X^t \mid X^{t-1}) > \mathrm{H}(\hat{X}^t \mid X^{t-1}) \tag{15}$$

As the conditional entropy is negatively correlated with the mutual information, the conditional entropy decreases after alignment, and the mutual information $\mathrm{MI}(X^{t-1}, X^t)$ increases correspondingly.

$$\mathrm{MI}(X^{t-1}, \hat{X}^t) > \mathrm{MI}(X^{t-1}, X^t) \tag{16}$$

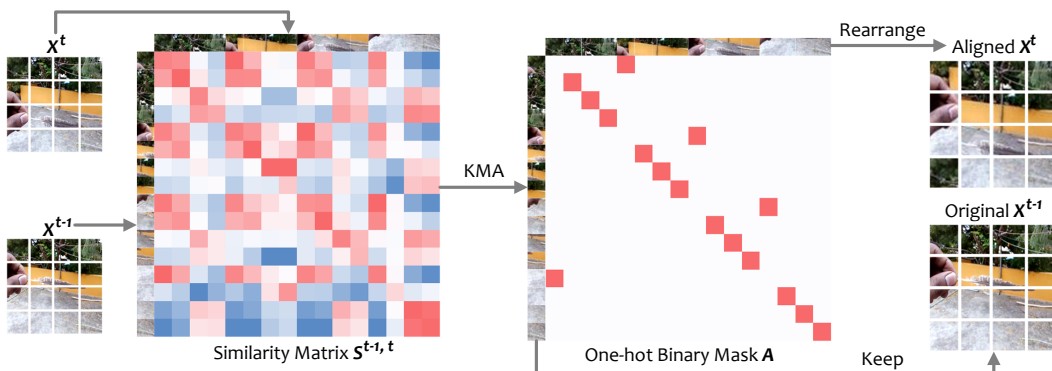

Figure 5: **Alignment with Kuhn-Munkres Algorithm (KMA).** Redder cells indicate higher similarities for paired features, while bluer ones mean the opposite.

The performance of a model is related to how much task-relevant information it collects from its input [45]. In video learning tasks, this task-relevant information is a subset of frame representations, which are information extracted from input frames. For simplicity, we present an instance of two neighboring frames demonstrated in Fig. 4. Under the same model capacity, information extracted from each frame is approximately the same amount for each model. Nevertheless, the situation is different in task-relevant information. When modeling with factorized (2D+1D) spatial-temporal operations in Fig. 4(a), the shared information of frame representations is rather small due to the misalignment of patches participating in a single temporal operation. With a more global spatial-temporal view, joint (3D) operations in Fig. 4(c) tend to increase this shared information. It also results in more task-relevant information extracted but is still restricted by the kernel/window size. Since enlarging kernel/window size cubically raises the complexity, this line of work is often less efficient. As for our alignment-guided spatial-temporal operation in Fig. 4(b), it can increase the mutual information shared by neighboring frame representations with simple operations, thereby having the potential to collect more task-relevant information at a low cost. This helps with both the sufficiency and the efficiency of cross-frame cross-location interactions and leads to better performance.

### 3.3 Solution

In order to solve the problem discussed, the key is to obtain an alignment matrix $A$. To this end, potential algorithms are supposed to produce a one-on-one matching matrix according to the given criterion. This can be simplified as a classical bipartite matching problem, which is efficient to solve through Kuhn-Munkres Algorithm (KMA) [4].

**Alignment.** Given the cosine similarity criterion as in Eq. (5), the alignment procedure with KMA is illustrated in Fig. 5. Firstly, we calculate the similarity matrix of two adjacent frames $X^{t-1}$ and $X^t$

$$S^{t-1,t} = \sum_{i=j} \frac{x_i^{t-1} \cdot x_j^t}{\|x_i^{t-1}\|\|x_j^t\|} \tag{17}$$

Based on this, KMA produces a one-hot binary matrix matching patch embeddings from the previous frame and patches from the current

$$A = \text{KMA}\left(S^{t-1,t}\right) \tag{18}$$

Then the spatial order of the previous frame features is kept, and that of the current is rearranged according to Eq. (3). This process is applied to every neighboring frame pair, thereby producing an aligned route for 1D temporal operations. Since the computational complexity of KMA is $O(N^3)$ [19], alignment with KMA has a complexity of $O(TH^3W^3)$.

**De-alignment.** After aligning frame representations with rearrangement, the spatial structures of them are modified. To keep the original location-based information and facilitate subsequent spatial operations, a recovery with de-alignment as in Eq. (7) is thus needed. As a result, the alignment, the temporal operation, and the de-alignment as a whole form a temporal interaction through a 3D path, within which each point on the path is similar to the next one.

Table 1: **Comparison to the state-of-the-art on Kinetics-400.** $x \times y$ views means $x$ spatial crops with $y$ temporal clips. We report the inference cost with a single view. **RED**/**BLUE** indicate SOTA/the second best. IN stands for officially released ImageNet [23] pretrained models. The same below.

| Method | Pretrain | Frames | Top-1 Acc | Top-5 Acc | Views | FLOPs (G) | Params (M) |
|---|---|---|---|---|---|---|---|
| ViT-B-VTN [33] | IN-21K | 16 | 79.8 | 94.2 | $1 \times 1$ | 4218 | 114 |
| VidTr-L [50] | IN-21K | 32 | 79.1 | 93.9 | $3 \times 10$ | 351 | - |
| ViViT-L [2] | IN-21K | 16 | 81.3 | 94.7 | $3 \times 4$ | 3992 | 310.8 |
| TimeSformer [3] | IN-21K | 8 | 78.0 | 93.7 | $3 \times 1$ | 196 | 121.4 |
| TimeSformer-L [3] | IN-21K | 96 | 80.7 | 94.7 | $3 \times 1$ | 2380 | 121.4 |
| TimeSformer-HR [3] | IN-21K | 16 | 79.7 | 94.4 | $3 \times 1$ | 1703 | 121.4 |
| MViTv1-B [11] | - | 16 | 78.4 | 93.5 | $1 \times 5$ | 70.5 | 36.6 |
| MViTv1-B [11] | - | 32 | 80.2 | 94.4 | $1 \times 5$ | 170 | 36.6 |
| MViTv1-B [11] | - | 64 | 81.2 | 95.1 | $3 \times 3$ | 455 | 36.6 |
| Mformer-B [34] | IN-21K | 16 | 79.7 | 94.2 | $3 \times 10$ | 369.5 | 109.1 |
| Mformer-L [34] | IN-21K | 32 | 80.2 | 94.8 | $3 \times 10$ | 1185.1 | 109.1 |
| Mformer-HR [34] | IN-21K | 16 | 81.1 | **95.2** | $3 \times 10$ | 958.8 | 381.9 |
| ATA (Ours) | IN-21K | 32 | **81.4** | **95.5** | $3 \times 1$ | 792.9 | 121.8 |
| ATA (Ours) | IN-21K | 32 | **81.9** | **95.5** | $3 \times 4$ | 792.9 | 121.8 |

Table 2: **Comparison to the state-of-the-art on Something-Something V2.** IN-21K+K-400 denotes pretraining the video architecture on K-400 based on ImageNet [23] pretrained image models.

| Method | Pretrain | Frames | Top-1 Acc | Top-5 Acc | Views | FLOPs (G) | Params (M) |
|---|---|---|---|---|---|---|---|
| ViViT-L [2] | IN-21K+K-400 | 16 | 65.4 | 89.8 | - | 903 | 352.1 |
| TimeSformer [3] | IN-21K | 8 | 59.5 | 74.9 | $3 \times 1$ | 196.7 | 121.4 |
| TimeSformer-L [3] | IN-21K | 96 | 62.4 | 81.0 | $3 \times 1$ | 2380 | 121.4 |
| TimeSformer-HR [3] | IN-21K | 16 | 62.2 | 78.0 | $3 \times 1$ | 1703 | 121.4 |
| MViTv1-B [11] | K-400 | 16 | 64.7 | 89.2 | $3 \times 1$ | 70.5 | 36.6 |
| MViTv1-B [11] | K-400 | 32 | 67.1 | 90.8 | $3 \times 1$ | 170 | 36.6 |
| MViTv1-B [11] | K-400 | 64 | **67.7** | 90.9 | $3 \times 1$ | 455 | 36.6 |
| Mformer-B [34] | IN-21K+K-400 | 16 | 66.5 | 90.1 | $3 \times 1$ | 369.5 | 109.1 |
| Mformer-L [34] | IN-21K+K-400 | 32 | **68.1** | **91.2** | $3 \times 1$ | 1185.1 | 109.1 |
| Mformer-HR [34] | IN-21K+K-400 | 16 | 67.1 | 90.6 | $3 \times 1$ | 958.8 | 381.9 |
| ATA (Ours) | IN-21K | 32 | 67.0 | **91.0** | $3 \times 1$ | 792.9 | 121.8 |
| ATA (Ours) | IN-21K | 32 | 67.1 | 90.8 | $3 \times 4$ | 792.9 | 121.8 |

# 4 Experiments

## 4.1 Experimental Setting

**Benchmarks.** We employ two widely used video action recognition datasets, i.e., Kinetics-400 (K-400) [22] and Something-Something V2 (SSv2) [14], in our experiments. Kinetics-400 contains 240k training videos and 30k validation videos in 400 classes of human actions. Something-Something V2 consists of 168.9K training videos and 24.7K validation videos for 174 classes. We provide the top-1 and top-5 accuracy on the validation sets, the inference complexity measured with FLOPs, and the model capacity in terms of the number of parameters.

**Implementation details.** We use TimeSformer [3] with the officially released model pretrained on ImageNet-21K [23] as our baseline. After inserting our ATA to each Transformer block, the model is then finetuned on video datasets with standard augmentations as in [12]. We adopt SGD to optimize our network for 30 epochs with a mini-batch size of 64. The initial learning rate is set to 0.005 with $0.1\times$ decays on the 21st and 27th epochs. All patch embeddings are applied with a weight decay of $1e-4$, while the class tokens and the positional embeddings used no weight decay. In the main experiments compared with the state-of-the-art methods, we provide the results of 32-frame input on both K-400 and SSv2. And in the ablation study, we provide the results of 8-frame input and $3\times1$ testing views on K-400. The resolution of $224 \times 224$ is used throughout all the experiments.

Table 3: **Ablation study of temporal modeling and de-alignment on Kinetics-400.** For temporal modeling, averaging refers to computing the mean of all spatial-temporal features, and attention means factorized 1D temporal attention. The same below.

| Image Backbone | Pretrain | Temporal Modeling | De-alignment | Top-1 Acc | Top-5 Acc | FLOPs (G) | Params (M) |
|---|---|---|---|---|---|---|---|
| CycleMLP-B5 [6] | IN-1K | Averaging [46] | - | 74.9 | 92.1 | 75.3 | 82.4 |
| | | Attention [3] | - | 76.8 | 93.1 | 122.4 | 102.8 |
| | | ATA (Ours) | ✗ | 72.7 | 90.9 | 122.4 | 102.8 |
| | | ATA (Ours) | ✓ | 77.7 | 93.5 | 122.4 | 102.8 |
| ConvNeXt-Base [31] | IN-22K | Averaging [46] | - | 79.0 | 94.0 | 123.0 | 88.0 |
| | | Attention [3] | - | 80.1 | 94.8 | 198.0 | 140.5 |
| | | ATA (Ours) | ✗ | 76.0 | 92.7 | 198.0 | 140.5 |
| | | ATA (Ours) | ✓ | 80.5 | 94.8 | 198.0 | 140.5 |
| ViT-Base [10] | IN-21K | Averaging [46] | - | 76.0 | 92.6 | 140.3 | 86.2 |
| | | Attention [3] | - | 78.0 | 93.7 | 201.8 | 121.8 |
| | | ATA (Ours) | ✗ | 79.3 | 94.3 | 201.8 | 121.8 |
| | | ATA (Ours) | ✓ | 79.6 | 94.3 | 201.8 | 121.8 |

## 4.2 Comparison Results

**Kinetics-400.** Table 1 presents comparison to the state-of-the-art results on K-400. As our motivation is to demonstrate the effectiveness of alignment without bells and whistles, we compare our model only with the approaches based on ViT to reduce the impact of architectural inconsistency. All these methods are pretrained on ImageNet-21K except for MViT [11], which is trained from scratch. Results show that our method outperforms most of the existing ViT-based approaches. It is noteworthy that our method has the same factorized spatial-temporal structure but a different form of temporal modeling with TimeSformer [3]. Using fewer input frames and lower resolution, ATA even surpasses TimeSformer-L and TimeSfomer-HR, which can fully demonstrate its effectiveness. Compared with Motionformer [34], which explores motion trajectory in temporal attention, our method achieves better performance with much lower complexity. Our approach can achieve better performance if using pretrained models on larger datasets, e.g., JFT-300M.

**Something-Something V2.** Table 2 compares our approach with the state-of-the-art methods on SSv2. Different from K-400, SSv2 consists of videos with high temporal reasoning. Therefore, pretraining on video data would be very helpful. That may be the reason why our method shows lower performance than Motionformer-L at the same input setting. It is noteworthy that our method achieves comparable results to MViT-B with 32-frame input. Considering its 3D structure and pretraining on K-400, our method demonstrates a stronger ability in temporal modeling.

## 4.3 Ablation study

**Generality on various image backbones.** To investigate the generality of ATA, we add three kinds of temporal modeling to three distinct image architectures (MLP-based, Conv-based, and attention-based) in Table 3. Note that we only consider ATA with de-alignment in this ablation. Based on each image backbone, temporal attention obtains improvements of 1.9%, 0.9%, and 2.0% accordingly compared with averaging, with the help of interactions among frames. Since the attention mechanism explicitly considers the correlation of frame-wise features, its outputs will contain more mutual information than the inputs. Moreover, with alignment and de-alignment, ATA achieves extra gains of 0.9%, 0.4%, and 1.6%. This is the result of better choices of attention participants, which further increases the mutual information among frames. Consequently, more mutual information gathered leads to a better understanding of video sequences, thereby bringing higher performance.

**Effectiveness of de-alignment.** We compare our ATA with and without de-alignment in the last two lines of each architecture in Table 3. For models based on the same image backbones, we see performance drops ranging from 0.3% to 5.0% in top-1 accuracy and from 0.0% to 2.6% in top-5 accuracy. This demonstrates the necessity of de-alignment. Moreover, we notice that models based on CycleMLP [6] (rows 1-2) and ConvNeXt [31] (rows 3-4) experience larger fall than that based on ViT [10] (rows 5-6). Its potential reason might be that CycleMLP [6] and ConvNeXt [31] rely heavier on local spatial features. These features will be confused by rearrangement-based alignment, and thus damage the learning of spatial structures. Since ViT [10] adopts the attention mechanism, which is global for spatial features, its spatial modeling is less affected.

Table 4: **Comparison of Mutual Information on Kinetics-400.** "None" denotes the result calculated right after patch embedding. Mutual information is averaged among all pairs of adjacent frames.

| Image Backbone | Pretrain | Temporal Modeling | Kinetics-400 | Something-Something V2 |
|---|---|---|---|---|
| ViT-Base [10] | IN-21K | None | 0.243 | 0.295 |
| | | Averaging [46] | 1.355 | 0.869 |
| | | Attention [3] | 1.371 | 0.910 |
| | | ATA (Ours) | 1.373 | 1.290 |

Table 5: **Ablation study of inserting locations on Kinetics-400.** Block $x$-$y$ means replacing the original temporal attention with our ATA on corresponding encoder blocks.

| Block 0-2 | Block 3-5 | Block 6-8 | Block 9-11 | Top-1 Acc | Top-5 Acc | FLOPs (G) | Params (M) |
|---|---|---|---|---|---|---|---|
| - | - | - | - | 78.0 | 93.7 | 201.8 | 121.8 |
| ✓ | - | - | - | 79.3 | 94.4 | 201.8 | 121.8 |
| - | ✓ | - | - | 79.6 | 94.4 | 201.8 | 121.8 |
| - | - | ✓ | - | 79.3 | 94.2 | 201.8 | 121.8 |
| - | - | - | ✓ | 79.2 | 94.2 | 201.8 | 121.8 |
| ✓ | ✓ | ✓ | ✓ | 79.6 | 94.3 | 201.8 | 121.8 |

**Comparison of mutual information.** As we theoretically proved, alignment increases mutual information between neighboring frame representations. With respect to this, practical results are in Table 4. Clearly, we can see that mutual information is extremely low without temporal modeling. Simple averaging can add to mutual information by 1.112 on K-400 and 0.574 on SSv2, while attention increases by 1.128 and 0.615 respectively. On the basis of attention, our parameter-free alignment obtains even more growth. We also notice that the difference that alignment makes is more pronounced on SSv2, possibly resulting from more dynamic temporal information on this dataset. This whole trend is correspondent to the performance. As a result, alignment can indeed increase mutual information and include more task-relevant information, leading to a soaring performance.

**Impact of inserting locations.** We divide 12 blocks of our baseline into 4 stages, each containing 3 blocks. Then we insert our ATA in each stage individually, as shown in Table 5. Generally, per-stage insertions of ATA in rows 2-5 obtain 1.2% $\sim$ 1.6% gains in top-1 accuracy and 0.5% $\sim$ 0.7% gains in top-5 accuracy compared with the baseline in the first row. These results are approximately the same as that in the last row, which equips every block in the baseline with alignment and de-alignment. This shows that even partially inserting ATA helps increase mutual information for adjacent frames, thereby benefiting the performance. We also notice that the 3rd row achieves the best performance in the table. It suggests that the selection of features to align and de-align also makes a difference, besides the operations themselves.

## 5   Limitation Discussion

As a preliminary attempt toward the frame alignment problem, ATA is both parameter-free and not differentiable, which might limit its upper-bound of performance. To this end, the process of matching features from adjacent frames can be potentially conducted by a network, or it can be modeled by consecutive displacements rather than a discrete reordering of features. These advances may enable end-to-end training of the model, leading to a more data-driven pipeline. Moreover, a soft version of alignment instead of a hard permutation can make this operation differentiable. This transforms the one-on-one linking between adjacent frames into a distribution-based pattern, which is more intuitive.

## 6   Conclusion

In this paper, we investigate temporal modeling approaches from the perspective of the sufficiency and the efficiency of spatial-temporal interactions. Through theoretical analysis, we find that increasing mutual information in neighboring frame representations can potentially promote task-relevant information collected by the model, and thus benefits its performance. Then we propose a general alignment and de-alignment scheme to enlarge the mutual information among consecutive frames. For specific implementation, we propose our Alignment-guided Temporal Attention (ATA) to conduct feature-level alignment and similarity-based temporal modeling. Extensive experiments show the effectiveness and generality of ATA. Our proposed module not only achieves satisfying results on multiple video action recognition benchmarks, but also can act as a general plug-in to extend any image backbone for video learning tasks.

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
