# Alignment-guided Temporal Attention for Video Action Recognition Supplemental Material

## A Implementation Details

Implementing alignment and de-alignment on the basis of the Kuhn-Munkres Algorithm (KMA) [2] is extremely simple. We provide their PyTorch-like pseudo-code in Algorithms 1 and 2.

---

**Algorithm 1** Alignment algorithm, PyTorch-like

---

```python
# Inputs:
# x: Features with shape [T, H, W, C]

# Outputs:
# x: Aligned features with shape [T, H, W, C]
# dealignment: Binary mask matrix for de-alignment with shape [T, HW, HW]

def Align(x):
    dealignment = []
    x = x.reshape(T, H * W, C) # [T, HW, C]

    for ti in range(1, T):
        prev = normalize(x[ti - 1], dim=-1).detach() # [HW, C]
        curr = normalize(x[ti], dim=-1).detach() # [HW, C]
        similarity = prev @ curr.t() # [HW, HW]
        # calculate the one-hot matching matrix with KMA
        alignment = linear_sum_assignment(similarity) # [HW, HW]
        dealignment.append(alignment.t())
        x[ti] = x[ti, alignment.argmax(-1)]

    dealignment = stack(dealignment) # [T, HW, HW]
    x = x.reshape(T, H, W, C) # [T, H, W, C]

    return x, dealignment
```

---

**Algorithm 2** De-alignment algorithm, PyTorch-like

---

```python
# Inputs:
# x: Aligned features with shape [T, H, W, C]
# dealignment: Binary mask matrix produced in alignment with shape [T, HW, HW]

# Outputs:
# x: De-aligned features with shape [T, H, W, C]

def Dealign(x, dealignment):
    x = x.reshape(T, H * W, C) # [T, HW, C]

    for ti in range(1, T):
        x[ti] = x[ti, dealignment[ti].argmax(-1)]

    x = x.reshape(T, H, W, C) # [T, H, W, C]

    return x
```

---

Table 1: **Ablation study of complexity on Kinetics-400.** $T$, $H$, $W$, and $d$ represent the temporal and spatial sizes of the input feature and the corresponding dimension. Complexities are calculated for each encoder block. We report results on the ViT-Base [3] backbone, with the same below.

| Temporal Modeling | Complexity | Top-1 Acc | Top-5 Acc | FLOPs (G) | Params (M) |
|---|---|---|---|---|---|
| Spatial Attention [1] | $O(TH^2W^2d)$ | 76.0 | 92.6 | 140.3 | 86.2 |
| Joint Attention [1] | $O(T^2H^2W^2d)$ | 77.4 | 92.8 | 179.7 | 86.2 |
| Factorized Attention [1] | $O(TH^2W^2d + T^2HWd)$ | 78.0 | 94.3 | 201.8 | 121.8 |
| ATA (Ours) | $O(TH^3W^3 + TH^2W^2d + T^2HWd)$ | 79.6 | 94.3 | 201.8 | 121.8 |

Table 2: **Ablation study of the number of alignment blocks on Kinetics-400.** Block $x$-$y$ means replacing the original temporal attention with our ATA on corresponding encoder blocks.

| Block 0-2 | Block 3-5 | Block 6-8 | Block 9-11 | Top-1 Acc | Top-5 Acc | FLOPs (G) | Param (M) |
|---|---|---|---|---|---|---|---|
| - | - | - | - | 78.0 | 93.7 | 201.8 | 121.8 |
| ✓ | - | - | - | 79.3 | 94.4 | 201.8 | 121.8 |
| ✓ | ✓ | - | - | 79.2 | 94.3 | 201.8 | 121.8 |
| ✓ | ✓ | ✓ | - | 79.4 | 94.4 | 201.8 | 121.8 |
| ✓ | ✓ | ✓ | ✓ | 79.6 | 94.3 | 201.8 | 121.8 |

## B   Ablation Study

**Complexity of KMA-based alignment.** As shown in Table 1, we analyze various temporal modeling approaches in terms of complexity. Since the complexity of the Kuhn-Munkres Algorithm [2] is only quadratic with $HW$ rather than with $THW$ as joint (3D) spatial-temporal attention, the complexity of our ATA keeps the same order as factorized attention while providing better results. Here we use an assumption, $O(HW) = O(d)$, which makes

$$O(TH^3W^3 + TH^2W^2d + T^2HWd) = O(TH^2W^2d + T^2HWd) < O(T^2H^2W^2d) \quad (1)$$

That is to say, when the size ($hw$) or number ($t$) of input frames grow, the computational overhead of ATA will increase slower than that of joint spatial-temporal attention. In practice, we find that joint spatial-temporal attention costs more memory since it attends to all patches in a single attention operation. Therefore, our ATA is more efficient than other attention-based methods, meanwhile bringing higher performance.

**Impact of the number of alignment blocks.** In Table 2, we explore the difference ATA makes when it is inserted into different numbers of encoder blocks. Generally, the models perform better with alignment compared with those without it. Top-1 and top-5 accuracy gradually grow with ATA added to more stages, with a maximum growth of 1.6%. The model benefits partially from aligning features at different depths, thus aligning features from disparate hyperspaces and increasing mutual information for every block of features. It is also worthwhile to notice that when ATA is added to only one single stage, the model gains 1.3% compared with plain temporal attention. This indicates that alignment is effective even if only added to a small range of temporal operations.

## C   Qualitative Analysis

To investigate the behaviors of three temporal modeling paradigms introduced in our main paper, we visualize their feature map responses in Fig. 1 and Fig. 2. The first row of each figure presents the original frames. Succeeding rows are final or intermediate heatmaps.

Looking into Fig. 1, all three modeling methods focus more on the foreground than on the background. Despite this, temporal modeling with simple averaging in row 2 obtains even and spread-out responses. Temporal attention in row 3 activates more intensely in the foreground compared with averaging. Nevertheless, confined to insufficient cross-frame cross-location interactions, the model tends to relate lose sight of inconsistent contents in the foreground, such as the head shown in $F_7$ and $F_8$. Our model in the last row addresses this issue through frame-by-frame alignment, thereby leading to a continued distinction between the activation of foreground and background.

We see a similar trend in Fig. 2. As a dataset emphasizes more on motion signals rather than appearance information, Something-Something V2 requires models to have a stronger ability in capturing temporal information. Comparing row 2 with row 3, we find that temporal attention leans toward highlighting similar contents among all frames, which sometimes can be the background. This is probably due to its cross-frame interaction pattern which only fuses a fixed spatial area. In contrast, our ATA concentrates more on moving objects, which are more likely to be clues of actions.

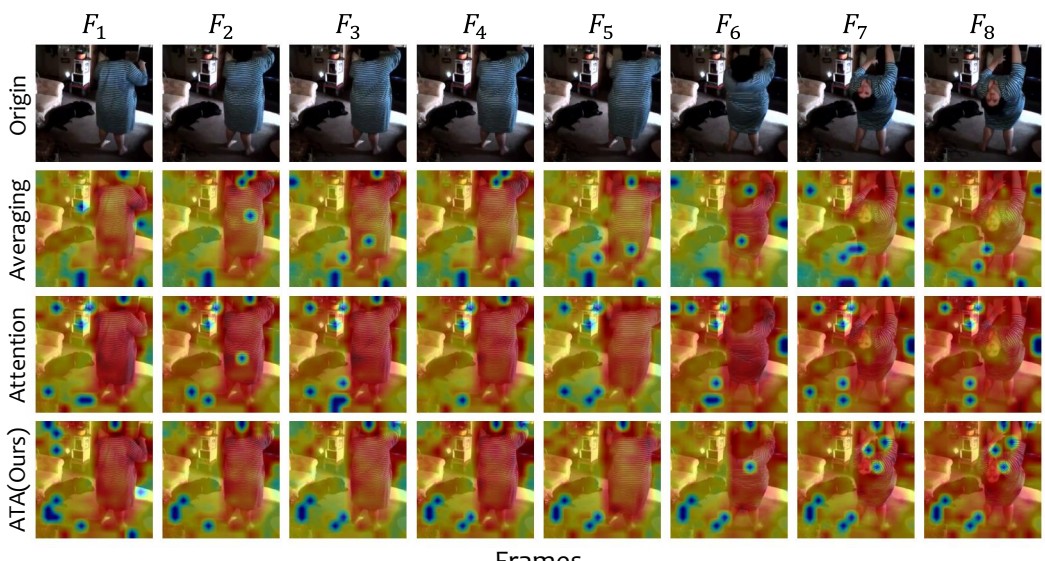

Figure 1: **Final feature map responses of different temporal modeling on Kinetics-400.** Rows 2-4 are produced by averaging all channels of the feature before the classification head of each model.

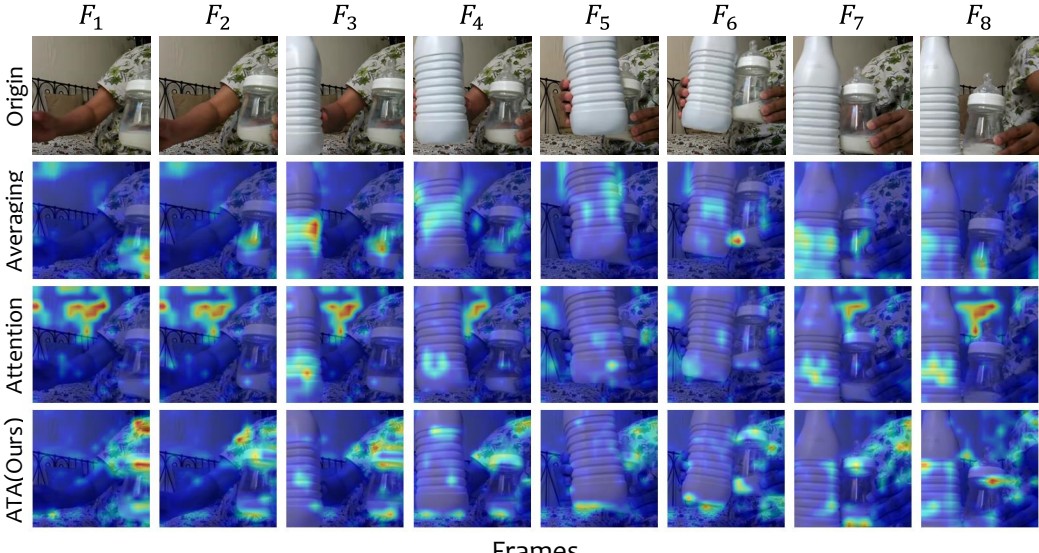

Figure 2: **Intermediate feature map responses of different temporal modeling on Something-Something V2.** Rows 2-4 are produced by averaging all channels of the feature after temporal modeling of the $7^{th}$ encoder block of each model.