# OpenReview forum: "Alignment-guided Temporal Attention for Video Action Recognition"
_NeurIPS.cc/2022/Conference — NeurIPS 2022 Accept_

### Official Review · Reviewer_Fgtq · 2022-07-02

**Rating:** 6
**Confidence:** 4
**Soundness:** 3 good
**Presentation:** 3 good
**Contribution:** 3 good

**Summary:**

This paper introduce a novel alignment-guided transformer architecture for video action recognition. Specifically, the proposed method makes use of KMA to obtain the alignment matrix. Then 1D temporal operation is further used to extract better video representation. The proposed method achieves competitive results on mutiple action recognition datasets.

**Questions:**

Q1.Have the reviewers investigates how the temporal sampling rate affects the proposed method?There are two ways to examine this.
1. If the input is temporally downsampled by x2 x4 x8, how does this affect the model performance
2. the ATA is done on every other frame, what is the performance difference

Q2. Does the model learn better motion representation? This can be done by conducting the arrow of time experiment as in [A,B]
[A] Xie, Saining, et al. "Rethinking spatiotemporal feature learning: Speed-accuracy trade-offs in video classification." Proceedings of the European conference on computer vision (ECCV). 2018.
[B] Zhou, Bolei, et al. "Temporal relational reasoning in videos." Proceedings of the European conference on computer vision (ECCV). 2018.

Q3. How does the global and local motion affects the method. According to the authors, the model performs better on ssthv2. Is this because the videos in ssthv2 have more global motion. It would be great to see some analysis on this.

**Limitations:**

1. This paper needs some additional analysis on what has been learned by the proposed method. See questions raised in Questions section.
2. This paper should better describe the efficiency part. Though the proposed method is parameter free, how ever it does incur quite some additional computational cost when comparing with Timesformer backbone as shown in Table2. The authors should better tune the tone to help readers understand the additional computational cost

**Strengths And Weaknesses:**

+the core idea of using KMA to align spatial information for temporal operation is very interesting and novel to my best knowledge.

+The paper is overall well-written.

+The mathematical derivation is solid, and the results are strong.

+The proposed method is robust and can achieve consistent performance improvement using different backbones.


-The analysis on what has been learned by the model is not thorough enough, see questions section

-some details should be presented with more clarity see limitation section

---

> ### Author Response · Authors · 2022-08-02
> **Response to Reviewer Fgtq**
>
> We really appreciate your valuable comments. Below please find our specific answers to the questions. We will remain committed to clarifying any further questions that may arise during the discussion period.
>
> **The effect of temporal sampling rates.** For video recognition task, the input frames are sampled with even intervals. When the sampling rate is high, there are more input frames. Then, in order to better learn the temporal reasoning, the more frames input, the richer the video information obtained by the model, and the better the model's performance. We conducted experiments on three different input numbers in the temporal dimension: 32, 16 and 8, as shown in the following table. As the results in our paper use the input of 32 frames, the input of 16 frames and 8 frames corresponds to the downsampling rates of x2 and x4, respectively. The results demonstrate that the model's performance improves as the number of input frames increases.
>
> | Downsampling Rate | Input Frames | Top-1 Acc | Top-5 Acc | FLOPs (G) | Params (M) |
> | :---: | :---: | :---: | :---: | :---: | :---: |
> | x1 | 32 | 81.4 | 95.5 | 792.9 | 121.8 |
> | x2 | 16 | 80.9 | 95.1 | 398.3 | 121.8 |
> | x4 | 8 | 79.6 | 94.3 | 201.8 | 121.8 |
>
> **More analysis on what ATA has learned.** This is really a good suggestion. We have provided more qualitative analysis in Section D of our supplementary material. Figure 6 and Figure 7 show the feature map responses of temporal attention and ATA on Kinetics-400 and SSv2, respectively. In Figure 6, the feature maps generated by temporal attention and ATA both pay more attention to the areas where the action occurs. However, temporal attention lacks the ability of relating the corresponding regions with inconsistent contents, such as the head shown in $F_7$ and $F_8$. This trend is similar in the results of SSv2 in Figure 7. This demonstrates that ATA has learned not only the motion regions but also their changing trends.
>
> **The effect of global and local motion.** SSv2 contains videos with more complex motions than Kinetics-400. Besides global motion, there are also other types such as fast motions and inconsistent motions. We think that is the main reason why the performance of ATA is better on SSv2 than Kinetics-400. Therefore, to our understanding, ATA could handle more motion types than temporal attention.
>
> **Additional computational costs compared with TimeSformer baseline.** More input frames will lead to more computational costs. Under the same conditions (views and input frames), there are no additional computational costs for ATA compared with TimeSformer.

---

### Official Review · Reviewer_Ppgh · 2022-07-09

**Rating:** 6
**Confidence:** 5
**Soundness:** 3 good
**Presentation:** 3 good
**Contribution:** 3 good

**Summary:**

The paper tackles the problem of modeling temporal context from the input frames for video action recognition task. It discusses the current limitations of existing approaches based on factorized and joint convolutions. As a result, it proposes to use frame-by-frame alignment to potentially increase mutual information of frame representations. The proposed Alignment-guided Temporal Attention (ATA) is studied on two widely used action recognition datasets, Kinetics400 and Something-Something V2 showing improvement over baselines models.

**Questions:**

1) Why there is an increase in computational complexity over baseline TimeSformer with 8 frames. Is it just using a larger number of input frames (32)?
2) Would be more beneficial to have a more detailed comparison and discussion with different attention methods like vanilla attention and some factorized and more efficient versions of attention.

**Limitations:**

I find the discussion of the limitations is not sufficient and detailed.

**Strengths And Weaknesses:**

**Strengths**:
1) The paper is well-written with nice and useful visualizations. The motivation for the proposed method is clear.
2) The discussion of the limitations of the current standard convolutional approaches for temporal modeling is thorough. The theoretical results help to better understand the internal benefits of the method.
3) Ablation study shows the improvement over simple averaging and another attention-based method.
4) Combination of ATA with TimrSformer achieves state-of-the-art results, outperforming even larger variants of TimeSformer (TimrSformer-L and TimrSformer-HR).

**Weaknesses**:
1) I would like to see more discussion and comparison with attention-based methods? How does frame-by-frame alignment help in comparison to global attention? The complexity of the Kuhn-Munkres Algorithm is also cubic as in global attention. So what are the benefits?

---

> ### Author Response · Authors · 2022-08-02
> **Response to Reviewer Ppgh**
>
> We really appreciate your valuable comments. Below please find our specific answers to the questions. We will remain committed to clarifying any further questions that may arise during the discussion period.
>
> **More discussion and comparison with attention-based methods.** This is really a good suggestion. Our theoretical analysis shows that neither factorized (2D+1D) attention nor joint (3D, or global) attention obtains sufficient task-relevant information, thereby suffering from a relatively low performance. In existing methods, factorized attention and join (3D, or global) attention have fully exploited, such as TimeSformer, ViViT and Swin. In most cases, the two kinds of attention show similar performance. The following table shows the comparison of spatial attention, joint attention, factorized attention and our approach on TimeSformer based on ViT-Base model. Compared with joint attention, our ATA still has superior performance. We will include more discussion and comparison in our updated version.
>
> **The complexity analysis.** The complexity of the Kuhn-Munkres Algorithm is only cubic with $HW$ rather than with $THW$ as joint (global) attention. This keeps the complexity of our ATA the same order as factorized attention while providing better results, as shown below. Also, when the size ($hw$) or number ($t$) of input frames grow, the computational overhead of ATA will increase more slowly than that of joint (global) attention. In practical, we find that joint (global) attention costs more memory since it takes all $THW$ patches for attention. Therefore, our ATA is more efficient than other attention-based methods, bringing higher performance. We will further clarify this in our updated version.
>
> | Image Backbone | Temporal Modeling | Top-1 Acc | Top-5 Acc | FLOPs (G) | Params (M) | Complexity |
> | :---: | :---: | :---: | :---: | :---: | :---: | :---: |
> | ViT-Base | Spatial Attention | 76.0 | 92.6 | 140.3 | 86.2 | $TH^3W^3$ |
> | ViT-Base | Joint Attention | 77.4 | 92.8 | 179.7 | 86.2 | $T^3H^3W^3$ |
> | ViT-Base | Factorized Attention | 78.0 | 94.3 | 201.8 | 121.8 | $TH^3W^3 + T^3HW$ |
> | ViT-Base | ATA (Ours) | 79.6 | 94.3 | 201.8 | 121.8 | $TH^3W^3 + T^3HW$ |
>
> **Computational complexity with different number of input frames.** Your understanding is correct. According to the complexity analysis, more input frames lead to higher FLOPs while the same number of parameters.

---

> > ### Comment · Reviewer_Ppgh · 2022-08-09
> > **Thank you**
> >
> > I appreciate the response from the authors who answer my questions. I have a favorable opinion of the paper and will keep my initial "Weak Accept" rating. I think it is a solid enough submission.

---

### Official Review · Reviewer_Shks · 2022-07-17

**Rating:** 4
**Confidence:** 5
**Soundness:** 2 fair
**Presentation:** 3 good
**Contribution:** 2 fair

**Summary:**

This paper presents a method for video action recognition. It discusses the problems of factorized (2D+1D) and joint (3D) spatial-temporal operations and proposes Alignment-guided Temporal Attention (ATA) for temporal modeling. ATA achieves strong performance while maintaining low computational overheads. What’s more, this paper theoretically proves that the frame-by-frame alignment increases mutual information between neighboring frame representations, thereby potentially including more task-relevant information. Extensive experiments show the effectiveness and generality of ATA, which not only achieves satisfying results on multiple video action recognition benchmarks, but also can act as a general plug-in to extend any image backbone for video learning tasks.

**Questions:**

1.	The idea of aligned temporal attention is reasonable, however, the alignment method seems to be not innovative enough. It seems that the pixel alignment method in this paper is as same as that in reference [4]. Could you please clarify the difference of these two alignments?
2.	Could you please explain more about the Figure 1 (a)? How can the Task & representations shared information (illustrated by red line) be obtained? Why is it a trapezoid? Is the Task-relevant information obtained from RGB videos?
3.	It seems that the paper doesn’t cite the SOTA results to make the comparison, for example the articles below should be cited and compared.
-	Liu Z, Ning J, Cao Y, et al. Video swin transformer. CVPR. 2022.
-	Li K, Wang Y, Gao P, et al. Uniformer: Unified transformer for efficient spatiotemporal representation learning. ICLR, 2022.
4.	When calculating the alignment with Kuhn-Munkres Algorithm, cosine similarity is employed. Why the cosine similarity is chosen? Could you give some ablation studies based on the choice of similarity?


**Limitations:**

1.	This paper indicates that the aligned temporal attention can mine more task-relevant information from the aligned pixels. However, the spatial-temporal self-attention structure can model the attention among all the pixels along the time axis. Consequently, it seems that the alignment doesn’t have much significance.
2.	It seems that the results don’t have a huge improvement. For example, in Table 1, there is no large difference between the results of MViTv1-B, ViViT-L, Mformer-HR and ATA for Top-1 Acc.


**Strengths And Weaknesses:**

Strengths:
The paper is well organized and the writing is clear. The figures are well-understood. The experiments seem to be limited but convincing.
Weaknesses:
This paper doesn’t propose a model which is original and innovative enough. Because the main contribution is the aligned temporal attention operation, however the alignment operation seems to be as same as the alignment method in reference [4].

---

> ### Author Response · Authors · 2022-08-02
> **Response to Reviewer Shks**
>
> We really appreciate your valuable comments. Below please find our specific answers to the questions. We will remain committed to clarifying any further questions that may arise during the discussion period.
>
> **Relationship with Reference 4.** We would like to clarify the novelty of our method. The primary goal of this work is to prove the effectiveness of temporal alignment, theoretically and practically, on existing video and image backbones without any model-specific design, rather than to propose specific models. In other words, we hope to find a generic plug-in temporal alignment module to assist existing models. The method from reference [4] is a classic algorithm for solving bipartite matching problem rather than alignment problem. We adopt it only as an interchangeable implementation for patch-wise feature matching between adjacent frames. Therefore, our approach borrows the implementation of KMA of reference [4] and solves a different problem. We will include this clarification in our updated version.
>
> **Further clarification of Figure 1.** Task & representation shared information is obtained from two separate processes: task-relevant information learning and representation information learning. The former is the information needed by a specific task, e.g., Video Recognition, which is obtained in the training process in the form of the aggregation of multiple frames. And the latter is the information extracted from a particular frame. In the training process of video learning, the network learns to embed more task-relevant information in frame-wise representations. However, it cannot be guaranteed that the two processes harmonize. In other words, the shared part of these two kinds of information is variable. That is why we illustrate it as a trapezoid shape. Our motivation is to increase this shared information, so the polygon with the red frame in Figure 1(a) expands to the rectangle in Figure 1(b).
>
> **Comparison with other SOTA results.** This is really a good question. In our paper, we only compared our approach with ViT-based architectures, because the gaps between different architectures, e.g., network structure and pretraining models, may affect the evaluation of our concept. Swin and Uniformer have superior performance and pretraining models to ViT under the same conditions, so that we may need to employ other tuning tricks in training process to fill such gaps. Therefore, we prefer to demonstrate the effectiveness of our approach separately by integrating it into different architectures in Table 4. However, we agree that we should include these SOTA results in our updated version to provide more information to readers.
>
> **Choice of similarity function.** We choose Cosine similarity in Kuhn-Munkres Algorithm since it has been widely used to measure the feature similarity in visual learning. However, the idea of ablating similarity function is interesting. We will be more than excited to conduct the ablation and include the results in our updated version.
>
> **Comparison with spatial-temporal self-attention.** Indeed, spatial-temporal (or joint) attention can model the relationship among all the pixels along the time axis. This would take more task-relevant information for representations of single frames. However, more redundant task-irrelevant information would also be introduced. According to the experiments of TimeSformer and ViViT, the spatial-temporal attention shows inferior performance compared to factorized (2D+1D) attention. Moreover, joint spatial-temporal attention suffers from high computational and memory costs. On the contrary, ATA shares the same complexity order with factorized attention while introducing more task-relevant information for performance improvement, which shows its potential.
>
> **Performance of ATA on Table 1.** In Table 1, ATA outperforms ViViT-L, MViT-B and Mformer-HR with small margins. However, ViViT-L has much more complexity than ATA; MViT-B employs joint attention structure with higher memory cost than ATA; and Mformer-HR employs high resolution input which also has more complexity than ATA. We will include more detailed analysis in the updated version.

---

### Official Review · Reviewer_1mVh · 2022-07-18

**Rating:** 4
**Confidence:** 4
**Soundness:** 3 good
**Presentation:** 3 good
**Contribution:** 2 fair

**Summary:**

This paper tackles the video action recognition task focusing on temporal modeling. An Alignment-guided Temporal Attention (ATA) module with similarity-based feature-level alignment is proposed to enlarge the mutual information among consecutive frames, and theoretical analysis shows that increasing mutual information in neighboring frames' feature representations can increase task-relevant information and benefit the model performance. Experiments are conducted on two video action recognition datasets, i.e., Kinetics-400 and Something-Something V2, and show that the proposed ATA module can be a general plug-in module to apply on any image backbones for video recognition task.




**Questions:**

1. For the results in Table 4, why are the results in the last line highlighted? The numbers in that line are not the best compared to others.

2. It is better to mark these symbols from equations (17) and (18) into the Figure 5 for easy understanding.

**Limitations:**

The authors mention the privacy issue of using these video action recognition models at certain circumstances, e.g. hotels and homes,
in the part of potential negative societal impacts. While the general solution of restricting the model usage permission is pointed, the specific policies still depend on multiple factors, e.g. social science, etc.




**Strengths And Weaknesses:**

- Strengths:
The theoretical analysis is a plus, showing that increasing mutual information in neighboring frames' feature representations can increase task-relevant information and bring performance improvement.

- Weaknesses:
1. the proposed ATA module contains one alignment component and one de-alignment component. Though it is motivated in L183 that “To keep the original location-based information and facilitate subsequent spatial operations, a recovery with de-alignment as in Eq. (7) is thus needed.", how will the de-alignment component affect the performance?

2. The main results shown in Table 1 and Table 2 do not show significant improvement compared to other models, without convincing justification for the relative inferior results. Some explanations like in L214 that"Our approach can achieve better performance if using pretrained models on larger datasets, e.g., JFT-300M."  just show the anticipation without the real result support.

3. For the ablation study of ATA module in different locations shown in Table 3, the ATA module is inserted in an increasing order. How about the independent performance of inserting ATA module into different locations?

---

> ### Author Response · Authors · 2022-08-02
> **Response to Reviewer 1mVh**
>
> We appreciate your valuable comments. Below please find our specific answers to the questions. We will remain committed to clarifying any further questions that may arise during the discussion period.
>
> **Ablation study for the de-alignment component.** We have conducted the ablation study for the de-alignment component on several mainstream video backbones in Table 6 of our supplementary material. We hereby summarize the results in the following table, which provides both performance and complexity comparisons between whether using de-alignment or not. According to the table, the de-alignment component brings performance improvements for all backbones at zero cost. We totally agree that this ablation will make our approach better understood. We will include this ablation in the updated main paper.
>
> | Image Backbone | Pretrain | De-alignment | Top-1 Acc | FLOPs (G) | Params (M) |
> | :---: | :---: | :---: | :---: | :---: | :---: |
> | CycleMLP-B5 | IN-1k | &cross; | 72.7 | 122.4 | 102.8 |
> | CycleMLP-B5 | IN-1k | &check; | 77.7 | 122.4 | 102.8 |
> | ConvNeXt-Base | IN-22k | &cross; | 76.0 | 198.0 | 140.5 |
> | ConvNeXt-Base | IN-22k | &check; | 80.5 | 198.0 | 140.5 |
> | ViT-Base | IN-1k | &cross; | 79.3 | 201.8 | 121.8 |
> | ViT-Base | IN-1k | &check; | 79.6 | 201.8 | 121.8 |
>
> **Justification of improvements.** In Table 1, the result of our ATA surpasses all other approaches on Kinect-400, even with relatively fewer temporal clips (views) and without high-resolution (HR) inputs. Furthermore, in Table 7 of our supplementary material, better results are achieved with more temporal clips, i.e., 81.9% top-1 accuracy. This leads to a 0.6% top-1 accuracy growth over the second best. In Table 2, our model obtains comparable results with others under the same inference setting (32 frames, 3 \times 1 views). The performance gap between our ATA and Mformer-L is mainly because it is using Kinetics-400, besides ImageNet, as pretraining data. Moreover, the primary goal of this work is to prove the effectiveness of temporal alignment on existing video and image backbones without any re-pretraining. Therefore, the reason why our approach sometimes shows marginal gain mainly lies in the gap between the performance of our baseline and other models. We agree that we should make more thorough discussion on the performance part to provide further insights. We will include these discussions in our revised manuscript.
>
> **Independent performance of ATA module in different locations.** We have investigated this in Table 5 of our supplementary material, showing that independently inserting ATA in encoder blocks helps increase mutual information for adjacent frames and benefits the performance. We summarize it in the following table. It can be observed that inserting ATA in early stage of the model achieves better performance improvement. Considering both Table 3 in the main paper and Table 5 in the supplementary material, we find that the selection of features to align and de-align also makes a difference, besides the operations themselves. We totally agree that this ablation study would provide further insights and will include this part in the updated main paper.
>
> | Block 0-2 | Block 3-5 | Block 6-8 | Block 9-11 | Top-1 Acc | FLOPs (G) | Params (M) |
> | :---: | :---: | :---: | :---: | :---: | :---: | :---: |
> | - | - | - | - | 78.0 | 201.8 | 121.8 |
> | &check; | - | - | - | 79.3 | 201.8 | 121.8 |
> | - | &check; | - | - | 79.6 | 201.8 | 121.8 |
> | - | - | &check; | - | 79.3 | 201.8 | 121.8 |
> | - | - | - | &check; | 79.2 | 201.8 | 121.8 |
> | &check; | &check; | &check; | &check; | 79.6 | 201.8 | 121.8 |
>
>
> **The highlighted results in Table 4.** The highlighted row shows the architecture we use in comparison with other state-of-the-art results, which is the same in other tables. We agree that the highlight in Table 4 somewhat leads to confusion and will remove it in our updated version.
>
> **Symbols in Figure 5.** We appreciate your recommendation and will improve our drawings accordingly.

---

### Meta-Review · Area_Chair_3AgF · 2022-08-23

**Recommendation:** Accept
**Confidence:** Less certain

**Metareview:**

Paper was reviewed by four reviewers, receiving: 2 x Borderline Rejects and 2 x Weak Accepts. Importantly, post rebuttal, [1mVh] mentioned upgrading the rating from Borderline Reject to Borderline Accept (though this is not reflected in final ratings). The general concerns raised by the reviewers included, limited improvements over the baselines and lack of certain ablations / comparisons. Much of these concerns have been addressed by various new experiments and discussions provided during the rebuttal period. [1mVh], [Ppgh] and [Fgtq] have all acknowledged that their concerns were largely resolved. [Shks], who remained the only negative reviewer, did not participate in the discussion nor acknowledged reading the author responses. AC has gone through the responses to comments of [Shks] and found them partially convincing. Overall, given the overall positive assessment of the reviewers and the generality of the proposed approach that can be combined with variety of architectures, the work will make a fine contribution to NeurIPS.

**Award:**

No

---

### Decision · Program_Chairs · 2022-09-14

Accept